# Coherently driven microcavity-polaritons and the question of superfluidity

R.T. Juggins [1], J. Keeling [2] & M.H. Szymańska[1]

Due to their driven-dissipative nature, photonic quantum fluids present new challenges in understanding superfluidity. Some associated effects have been observed, and notably the report of nearly dissipationless flow for coherently driven microcavity-polaritons was taken as a smoking gun for superflow. Here, we show that the superfluid response—the difference between responses to longitudinal and transverse forces—is zero for coherently driven polaritons. This is a consequence of the gapped excitation spectrum caused by external phase locking. Furthermore, while a normal component exists at finite pump momentum, the remainder forms a rigid state that is unresponsive to either longitudinal or transverse perturbations. Interestingly, the total response almost vanishes when the real part of the excitation spectrum has a linear dispersion, which was the regime investigated experimentally. This suggests that the observed suppression of scattering should be interpreted as a sign of this new rigid state and not a superfluid.

[1] Department of Physics and Astronomy, University College London, Gower Street, London WC1E 6BT, UK. [2] SUPA, School of Physics and Astronomy, University of St Andrews, St Andrews KY16 9SS, UK. Correspondence and requests for materials should be addressed to R.T.J. (email: richard.juggins@gmail.com) or to M.H.S. (email: m.szymanska@ucl.ac.uk)

O ne of the most spectacular emergent effects in quantum physics is that of superfluidity, which was first observed as a set of peculiar flow properties in liquid helium when cooled below 2.17 K[1]. Understanding collective effects such as dissipationless flow, lack of response to transverse perturbations, quantised vortices, and metastable persistent currents has been the aim of much theoretical work[2–6], particularly with respect to systems in thermodynamic equilibrium. Extending these well-established ideas to driven-dissipative systems, which do not thermalise due to constant pumping and decay, has however proved contentious[7–13].

While dissipationless flow, explained by the Landau criterion[14], is perhaps the most famous property of superfluids, arguably the most fundamental is that they do not respond to transverse perturbations[5]: that is, the bulk of the fluid is irrotational. Crucially, this difference between longitudinal and transverse response onsets sharply at the phase transition, whereas perfectly dissipationless flow only occurs at absolute zero temperature, where the normal component vanishes. Thus, to clearly distinguish a true superfluid from a fluid merely with low viscosity, one needs to focus on the response functions. This difference in response derives from the dependence of superflow on the gradient of the macroscopic wavefunction phase, a fact that also leads to quantised circulation and the existence of vortices and persistent currents[2]. Experimentally, the absence of transverse response is striking with a well-known manifestation being the Hess-Fairbank effect (analogous to the Meissner effect in superconductors). Following this logic, the standard definition of the superfluid fraction is given by finding what part of the system responds to longitudinal, but not transverse, perturbations[3–6, 9,15–17]. Such a definition of superfluidity is equivalent to the use of the Meissner effect to distinguish a superconductor from a material with low resistance.

Driven-dissipative systems present new challenges in the study of superfluidity as they do not usually thermalise and it is unclear whether the effects seen in equilibrium will all continue to apply[9–11,15,18,19]. Examples of such systems are numerous, including Bose–Einstein condensates of photons[20,21], cold atoms coupled to photonic modes in optical cavities[22], and cavity arrays[23–25]. Much recent research in this area has been focused on microcavity-polaritons[15,26,27], which are bosonic quasiparticles made of quantum well excitons strongly coupled to cavity photons (see Fig. 1). Polariton experiments have observed a number of effects usually associated with superfluidity, such as the suppression of scattering for flow past a defect[28–30], quantised vortices[7], and metastable persistent currents[31].

The way a polaritonic system is pumped affects its excitation spectrum and so is likely to alter its superfluid properties. While experimentally uninvestigated, the transverse response of incoherently pumped polaritons has been calculated using a Keldysh path integral method[9,13], and it was found that a finite superfluid fraction can exist despite the system being out-of-equilibrium. Fundamentally, this was a direct consequence of the gaplessness of the diffusive excitation spectrum[32]. This conclusion seems to hold even when going beyond a linearised theory, and considering the full nonlinear dynamics which lead to a Kardar–Parisi–Zhang (KPZ) equation predicting the absence of analgebraic order[33]. Superfluidity is endangered by repulsive forces between vortices in the KPZ-ordered phase at large distances[12,34]—however, this can only be seen at scales too large to be relevant in current experiments.

On the other hand, coherently driven polaritons below the threshold of optical parametric oscillation (OPO), which we consider in this work, are quite different. Here, the system inherits its macroscopic phase from the pump, resulting in a gapped excitation spectrum[35]. This phase fixing also inhibits superfluid effects such as the formation of vortices and solitons. However, it is notable that nearly dissipationless flow has been observed in this system and described as evidence of superfluidity[28].

In this article, we calculate the longitudinal and transverse response functions for coherently pumped polaritons below the OPO threshold. We consider the case of a continuous and homogeneous pump—a regime in which dissipationless flow was reported in experiment[28]. We find these response functions to be equal, meaning that no part of the system responds to perturbations like a superfluid. Furthermore, we discover that a fraction of the system cannot be classified as 'normal' or 'superfluid' as it does not respond to either longitudinal or transverse forces. We show this rigidity to follow from external phase fixing. Because the excitation spectrum is gapped, the KPZ nonlinearity is not relevant and our result holds in the thermodynamic limit[13,33].

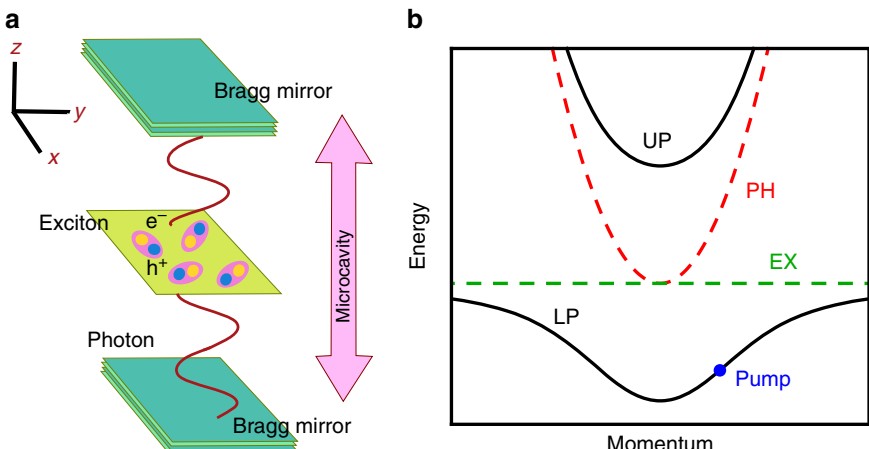

**Fig. 1** Polaritons in semiconductor microcavities. **a** Polaritons are quasiparticles formed when cavity photons, which are massive due to confinement in the z direction between two Bragg mirrors, interact strongly with excitons confined in a quantum well. Polaritons are free to move in the two-dimensional plane perpendicular to their confinement. **b** The excitonic dispersion (dashed green) is approximately constant compared to the photonic (dashed red) due to the much larger exciton mass. Strong coupling leads to anticrossing and the formation of upper and lower polariton branches (solid black). Polaritons interact because of their excitonic component, while their photonic part causes decay and the need for an external drive. A coherent laser pump resonantly tuned to the polariton dispersion is marked by a blue dot

## Results

**Gapped excitation spectrum.** We study microcavity-polaritons, which are two-dimensional bosonic quasiparticles that result from the strong coupling of quantum well excitons to cavity photons (see Fig. 1). They have a very small mass ($\sim 10^{-4}\, m_e$, where $m_e$ is the mass of an electron) which allows the formation of macroscopically ordered states at high temperatures ($\sim 20$ K in GaAs[36], and up to room temperature in organic materials[37]).

More specifically, we are interested in coherently pumped microcavities, where laser excitation resonant or near-resonant with the lower polariton dispersion maintains a steady state against polariton decay, leading to the formation of a macroscopically occupied state at the pump frequency and momentum ($\omega_p$ and $\mathbf{k}_p = (k_p, 0)$)[38]. In such systems, there is no spontaneous symmetry breaking as the phase is fixed by the pump.

As we are interested in low energies, we can ignore the upper polariton dispersion and write the Hamiltonian of the system in terms of lower polariton operators, $\hat{a}$, coupled to bosonic decay bath modes, $\hat{A}$[39]:

$$
\begin{aligned}
\hat{H} = &\sum_{\mathbf{k}} \varepsilon_{\mathbf{k}} \hat{a}_{\mathbf{k}}^{\dagger} \hat{a}_{\mathbf{k}} + \frac{F_p}{\sqrt{2}} \left( \hat{a}_{\mathbf{0}}^{\dagger} + \hat{a}_{\mathbf{0}} \right) \\
&+ \frac{V}{2} \sum_{\mathbf{k},\mathbf{k}',\mathbf{q}} \hat{a}_{\mathbf{k}-\mathbf{q}}^{\dagger} \hat{a}_{\mathbf{k}'+\mathbf{q}}^{\dagger} \hat{a}_{\mathbf{k}} \hat{a}_{\mathbf{k}'} + \sum_{\mathbf{p}} \omega_{\mathbf{p}}^{A} \hat{A}_{\mathbf{p}}^{\dagger} \hat{A}_{\mathbf{p}} \\
&+ \sum_{\mathbf{k},\mathbf{p}} \zeta_{\mathbf{k},\mathbf{p}} \left( \hat{a}_{\mathbf{k}}^{\dagger} \hat{A}_{\mathbf{p}} + \hat{A}_{\mathbf{p}}^{\dagger} \hat{a}_{\mathbf{k}} \right),
\end{aligned} \tag{1}
$$

where momentum arguments are with respect to the pump frame. As a result, the lower polariton dispersion takes the form $\varepsilon_{\mathbf{k}} = (\mathbf{k} + \mathbf{k}_p)^2 / 2\, m^*$, where we have used a quadratic approximation. Other quantities appearing in this expression are: $F_p$, the amplitude of the pump, $V$, the polariton–polariton interaction strength, $\omega_{\mathbf{p}}^A$ the dispersion of the bath modes, and $\zeta_{\mathbf{k},\mathbf{p}}$, the coupling between the polaritons and the bath modes. While for

some range of detunings and pump intensities this system becomes bistable, here we consider the monostable state at low pump intensity.

The excitation spectrum of this system has been studied previously[35,39,40], and is given by

$$
\omega_{\mathbf{k}}^{*,\pm} = \frac{\alpha_{\mathbf{k}}^{+} - \alpha_{\mathbf{k}}^{-}}{2} - i\kappa \pm \frac{1}{2} \sqrt{\left( \alpha_{\mathbf{k}}^{+} + \alpha_{\mathbf{k}}^{-} \right)^2 - 4 V^2 |\psi_0|^4}, \tag{2}
$$

where $\psi_0$ is the mean-field solution, $\alpha_{\mathbf{k}}^{\pm} = \varepsilon_{\pm\mathbf{k}} - \omega_p + 2V|\psi_0|^2$, and $\kappa$ is a decay constant derived from integrating out the bath. Here, we wish to emphasise a point somewhat neglected in previous work, which has concentrated on the gaplessness of the real part of the excitation spectrum for specific blue detuning ($\Delta_p = \omega_p - \varepsilon_0 - V|\psi_0|^2 = 0$), at which it takes a linear Bogoliubov-like form near $\omega, k = 0$. It has been noted that in this regime, the real part appears to fulfil the Landau criterion for superfluidity, and this fact has been used to explain the observation of dissipationless flow[28]. However, we note that unlike in equilibrium systems, for which the criterion was derived, the excitation spectrum here is complex and, due to phase fixing by the pump, is gapped, except at exactly the pump strength where a parametric instability first occurs. In general this gap is found in the imaginary part (see Fig. 2). In fact, it has been shown that scattering in these systems can only be reduced, not completely eliminated[19]. Furthermore, while approximately dissipationless flow can be explained by the real part of the excitation spectrum, a gapped spectrum may have important consequences for superfluidity more generally. Indeed, the limits of using the Landau criterion alone to interpret superfluidity in driven-dissipative systems can be seen from the incoherent case, where the diffusive excitation spectrum[32] does not fulfil it at all. In this context, a new generalised criterion in terms of the complex wave vector was formulated to explain dissipationless flow[8].

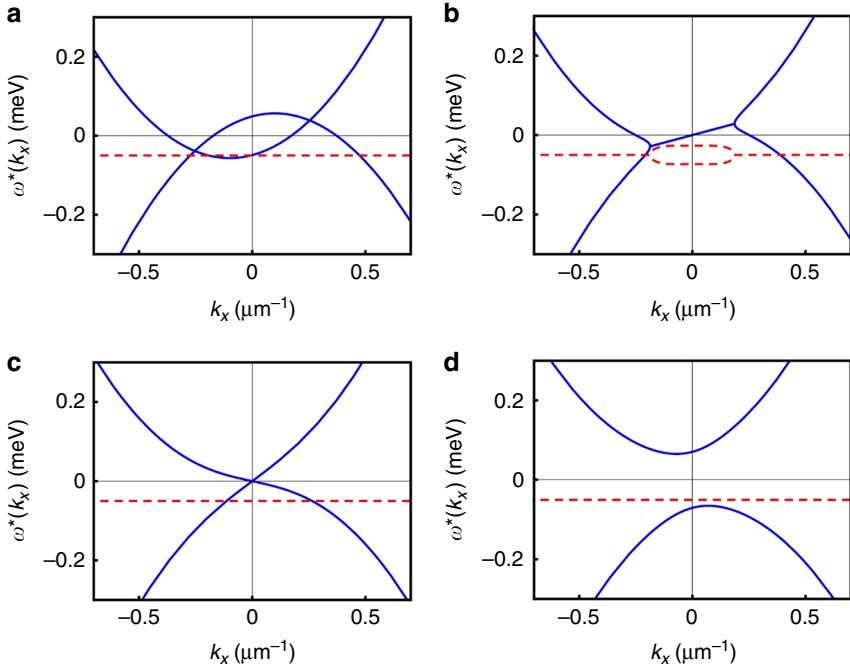

**Fig. 2** The excitation spectrum at different polariton densities. The real (solid blue) and the imaginary (dashed red) parts of the spectrum in the frame of the pump momentum, which is $k_p = 0.1\, \mu m^{-1}$ in the $x$-direction, for $k_y = 0$. The pump is blue detuned by 0.05 meV with respect to the bare lower polariton dispersion. In **a** and **b**, the densities are $|\psi_0|^2 = 0.2\, \mu m^{-2}$ and $|\psi_0|^2 = 9.3\, \mu m^{-2}$, respectively. In **c**, the density is $|\psi_0|^2 = 20.0\, \mu m^{-2}$ and the pump comes into resonance with the interaction shifted lower polariton dispersion, $\Delta_p = \omega_p - \varepsilon_0 - V|\psi_0|^2 = 0$, at which point the real part takes the linear form of the Bogoliubov spectrum. Here, and in **d** where $|\psi_0|^2 = 30.9\, \mu m^{-2}$, the Landau criterion is fulfilled in the real part. However, it is significant that the imaginary part is always gapped

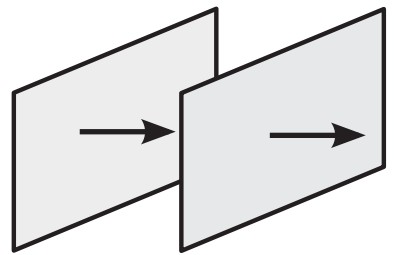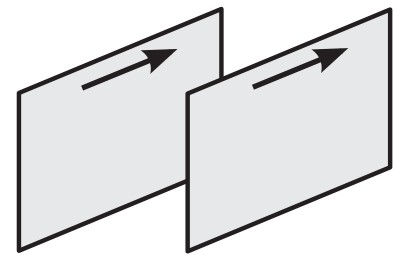

**Fig. 3** Longitudinal vs. transverse response. A current flowing in response to a perturbing force may originate through longitudinal or transverse actions, corresponding to pushing and shearing, respectively. A superfluid responds to longitudinal but not transverse perturbations[3–6,9,15–17]

**Sum-rules and anisotropy.** To study whether the phase fixing and gapped spectrum in coherently pumped polaritons affects their superfluid properties, we calculate the static current–current response function. More specifically, for a perturbation described by the Hamiltonian $\hat{H} = \sum_{\mathbf{q}} \hat{\mathbf{j}}(\mathbf{q}).\mathbf{f}(\mathbf{q})$, where $\mathbf{f}$ is the perturbing force, this response function describes the tendency for a particle current to flow as a result of that force, $j_i(\mathbf{q}) = \chi_{ij}(\mathbf{q})f_j(\mathbf{q})$, where $i$, $j$ refer to directions in the $xy$ plane. This function is given by the correlator of two current operators[41], $\chi_{ij}(\mathbf{q}) = -i\langle \hat{j}_i(\mathbf{q})\hat{j}_j(-\mathbf{q})\rangle$, and can be separated into longitudinal and transverse parts (i.e. the response to pushing or shearing the polaritons as shown in Fig. 3). A perfect superfluid has only the former. Mathematically, this decomposition can be made by considering a diagonal component of the response tensor, $\chi_{ii}$, and varying the order in which the components of the momentum vector are taken to zero[3,42]. Taking the transverse momentum (i.e. the component perpendicular to $i$) to zero before the longitudinal momentum (component parallel to $i$) gives the longitudinal response function, and vice versa.

In systems where particle number is conserved, the f-sum rule holds[3]. This identity follows directly from continuity of current,

$$[\hat{H}, \hat{\rho}(\mathbf{q})] = -\mathbf{q}.\hat{\mathbf{j}}(\mathbf{q}), \qquad (3)$$

and allows the longitudinal response function in the long wavelength limit to be identified with the total density, $\rho = m\chi^{\mathrm{L}}(\mathbf{q} \to \mathbf{0})$. In addition, using linear response theory, one can identify the normal density as the transverse response function in the same limit[3,6], $\rho_{\mathrm{n}} = m\chi^{\mathrm{T}}(\mathbf{q} \to \mathbf{0})$, and calculate the superfluid density as the difference between the two limiting responses[3–6,9], $\rho_{\mathrm{s}} = m(\chi^{\mathrm{L}}(\mathbf{q} \to \mathbf{0}) - \chi^{\mathrm{T}}(\mathbf{q} \to \mathbf{0}))$. In driven-dissipative systems, however, particle conservation does not hold and one should not expect the f-sum rule or, by extension the above expression for $\rho_{\mathrm{s}}$, to hold either. While previous work on incoherently pumped polaritons nevertheless found that the f-sum rule did hold[9], coherently pumped systems face further complications. In particular, for the total density of the system to equal the sum of the normal and superfluid components, $\rho = \rho_{\mathrm{n}} + \rho_{\mathrm{s}}$, the system must be Galilean invariant[2]. It is clear that Galilean invariance is broken for a coherently pumped system, as polaritons are injected at a fixed momentum, picking out a special frame of reference through the pumping term in the Hamiltonian (Eq. (1)). In addition, coherently pumped polaritons are anisotropic, meaning that the longitudinal and transverse response functions are tensors, not scalars. Given these subtleties, a relation between the response functions and densities is unknown. Thus, instead, we determine whether any part of the system responds to perturbations in a manner characteristic of a superfluid, as defined by the superfluid response:

$$\lim_{\mathbf{q} \to \mathbf{0}} \left( \chi_{ij}^{\mathrm{S}}(\mathbf{q}) \right) = \lim_{\mathbf{q} \to \mathbf{0}} \left( \chi_{ij}^{\mathrm{L}}(\mathbf{q}) - \chi_{ij}^{\mathrm{T}}(\mathbf{q}) \right), \qquad (4)$$

where $\chi_{ij}^{\mathrm{L}}$ and $\chi_{ij}^{\mathrm{T}}$ are the (anisotropic) longitudinal and transverse

response functions. We will not, however, associate these with densities.

**Response function.** In order to calculate the static current–current response function, we utilise a path integral method where the perturbation is modelled by source fields[9]. Here, we present the results of this calculation, with details of the derivation using Keldysh field theory given in the Methods.

Each term in the response function contains two factors of the momentum vertex,

$$\boldsymbol{\gamma}(\mathbf{k} + \mathbf{q}, \mathbf{k}) = \frac{1}{2m^*} \begin{pmatrix} 2k_{\mathrm{p}} + 2k_x + q_x \\ 2k_y + q_y \end{pmatrix}, \qquad (5)$$

which describes the bare coupling between the source fields and excitations carrying momentum $\mathbf{q}$. Because the pump breaks rotational invariance, the vertex is anisotropic. Without loss of generality, we consider the pump wavevector to be in the $x$ direction.

The full response can be written in terms of two components, one due to the mean-field and the other due to fluctuations:

$$\chi_{ij}(\mathbf{q}) = \chi_{ij}^{\mathrm{mf}}(\mathbf{q}) + \chi_{ij}^{\mathrm{fl}}(\mathbf{q}). \qquad (6)$$

The mean-field term is,

$$\chi_{ij}^{\mathrm{mf}}(\mathbf{q}) = \sum_{\sigma,\sigma' \in \pm} c_{\sigma,\sigma'}^{\mathrm{mf}}(\mathbf{q}) \gamma_i(\sigma\mathbf{q}) \gamma_j(\sigma'\mathbf{q}), \qquad (7)$$

where the coefficient $c_{\sigma,\sigma'}^{\mathrm{mf}}(\mathbf{q})$ is given in Supplementary Note 1, and the fluctuations term is:

$$\chi_{ij}^{\mathrm{fl}}(\mathbf{q}) = \frac{\mathrm{i}}{4} \int \frac{d^2\mathbf{k}}{(2\pi)^2} \left\{ \mathrm{iTr}[\mathcal{M}_{ij}^{\mathbf{q}}] + \int \frac{d\omega}{2\pi} \left( \mathrm{Tr}[\mathcal{D}_{\omega,\mathbf{k}}^{\mathrm{K}} \mathcal{M}_{ij}^{\mathbf{q}}] \right. \right.$$
$$\left. \left. - \mathrm{Tr}\left[ \mathcal{D}_{\omega,\mathbf{k}+\mathbf{q}}^{\mathrm{R}} \mathcal{A}_i^{\mathbf{k}+\mathbf{q},\mathbf{k}} \mathcal{D}_{\omega,\mathbf{k}}^{\mathrm{K}} \mathcal{B}_j^{\mathbf{k},\mathbf{k}+\mathbf{q}} \right. \right. \right.$$
$$\left. \left. \left. + \mathcal{D}_{\omega,\mathbf{k}+\mathbf{q}}^{\mathrm{K}} \mathcal{A}_i^{\mathbf{k}+\mathbf{q},\mathbf{k}} \mathcal{D}_{\omega,\mathbf{k}}^{\mathrm{A}} \mathcal{B}_j^{\mathbf{k},\mathbf{k}+\mathbf{q}} \right] \right) \right\}, \qquad (8)$$

where the retarded, advanced, and Keldysh Green's functions are defined by $\mathcal{D}^{\mathrm{R}} = -\mathrm{i}\theta(t - t')\langle[\Psi(t), \Psi^\dagger(t')]\rangle$, $\mathcal{D}^{\mathrm{A}} = \mathrm{i}\theta(t' - t)\langle[\Psi(t), \Psi^\dagger(t')]\rangle$, and $\mathcal{D}^{\mathrm{K}} = -\mathrm{i}\langle\{\Psi(t), \Psi^\dagger(t')\}\rangle$. The Green's functions, as well as $\mathcal{A}_i^{\mathbf{k}+\mathbf{q},\mathbf{k}}$, $\mathcal{B}_i^{\mathbf{k},\mathbf{k}+\mathbf{q}}$, and $\mathcal{M}_{ij}^{\mathbf{q}}$, are $2 \times 2$ matrices in Nambu space (i.e., the space of particle creation and annihilation operators). The matrices consist of a series of

different combinations of momentum vertices:

$$\mathcal{A}_i^{\mathbf{k+q,k}}/\mathcal{B}_i^{\mathbf{k+q,k}} = \tfrac{1}{2}(1+\hat{\sigma}_z)\gamma_i(\mathbf{k+q,k})$$
$$+ \tfrac{1}{2}(1-\hat{\sigma}_z)\gamma_i(-\mathbf{k-q,-k}) \qquad (9)$$
$$+ \sum_{\sigma \in \pm} \mathcal{C}_\sigma^{\mathcal{A}/\mathcal{B}}(\mathbf{q})\gamma_i(\sigma\mathbf{q}),$$

$$\mathcal{M}_{ij}^{\mathbf{q}} = \sum_{\sigma,\sigma' \in \pm} \mathcal{C}_{\sigma,\sigma'}^{\mathcal{M}}(\mathbf{q})\gamma_i(\sigma\mathbf{q})\gamma_j(\sigma'\mathbf{q}), \qquad (10)$$

where $\mathcal{C}_\sigma^{\mathcal{A}/\mathcal{B}}(\mathbf{q})$ and $\mathcal{C}_{\sigma,\sigma'}^{\mathcal{M}}(\mathbf{q})$ are $2 \times 2$ matrices in Nambu space, and $\hat{\sigma}_z$ is the third Pauli matrix. These coefficients are given in Supplementary Note 1.

As discussed further below, it is important to observe that each of these terms involves $1/det[(D^R)^{-1}(\omega=0, \mathbf{q})]$. One may also note that the total response function can be divided into two sets of contributions. Those involving the response of the condensate appearing through the coupling $\gamma(\mathbf{0,q}) = (2\mathbf{k}_p + \mathbf{q})/2\,m^*$, and those involving coupling to excitations through $\gamma(\mathbf{k+q,k}) = (2\mathbf{k}_p + 2\mathbf{k} + \mathbf{q})/2\,m^*$.

**Superfluid response.** To quantify the superfluid behaviour of the system, we need to take the long wavelength limit of the response function. In order for the longitudinal and transverse responses to differ—that is, for the order in which $q_x$ and $q_y$ are taken to zero to matter in any way—there needs to be singular behaviour as $\mathbf{q} \to \mathbf{0}$. Each term in Eq. (6) takes the ultimate form

$$\frac{h(\mathbf{q})}{det[(D^R)^{-1}(\omega=0,\mathbf{q})]}, \qquad (11)$$

where $h(\mathbf{q})$ is some polynomial in $\mathbf{q}$. Singular behaviour requires the denominator of this expression, the static inverse retarded Green's function, to vanish as $\mathbf{q} \to \mathbf{0}$. One may see that the requirement for this to happen is directly related to the gaplessness of the excitation spectrum. The spectrum, $\omega^*(\mathbf{k})$, in Eq. (2) is defined by

$$det[(D^R)^{-1}(\omega^*(\mathbf{k}), \mathbf{k})] = 0. \qquad (12)$$

If the spectrum is gapless, i.e. if $\omega^*(\mathbf{k} \to \mathbf{0}) = 0$, then the static inverse retarded Green's function will vanish as $\mathbf{q}$ goes to zero, and there can be a singular dependence of the response function on momentum. As the excitation spectrum of coherently pumped polaritons is always gapped, this instead gives a finite value for the static inverse retarded Green's function, so there cannot be any singular terms. Consequently, the longitudinal and transverse response functions are equal and Eq. (4) shows that no part of the system responds to perturbations like a superfluid.

**Rigid state.** In fact our result is more profound. In the case where $\mathbf{k}_p = \mathbf{0}$, the response function is exactly zero in the long wavelength limit: that is, the system does not respond to either longitudinal or transverse perturbations. This implies that the coherently pumped system forms a rigid state more akin to a solid than a fluid in that it has a finite density but shows zero response to small perturbations. Indeed, while it is already known that phase fixing prevents the formation of vortices[43], we can see that its effect is even more severe in its restriction of the response of the system. (If the pump is restricted in time or in space, then there can be free phase evolution in those times or places without a pump, allowing vortex formation to be recovered[44,45].)

When the pump is at non-zero wavevector, $\mathbf{k}_p \neq \mathbf{0}$, the behaviour is different. While the state remains rigid to

perturbations (in the sense that its momentum continues to be locked to the pump) the net current can change, by modifying the occupation of the state at $\mathbf{k}_p$: i.e. if the coherent state has a finite momentum, a change in the occupation of that state will change the total current, allowing a non-zero response to the applied force. Mathematically, this occurs because at finite $\mathbf{k}_p$, the perturbation couples to the pump state through the momentum vertex $\gamma(\mathbf{0,0}) = \mathbf{k}_p/2\,m^*$, and so a force can change the amplitude of the state at $\mathbf{k}_p$. This change manifests as a finite normal response with quadratic dependence on $k_p$. (By 'normal' we mean that the longitudinal and transverse responses are equal.) As a result, the mean-field part of the response, which is orders of magnitude larger than the part due to fluctuations, takes the form

$$\chi_{ij}^{(0)}(\mathbf{q} \to \mathbf{0}) = -\delta_{xi}\delta_{xj}\frac{k_p^2|\psi_0|^2(\bar{\psi}_0 + \psi_0)}{m^{*2}\left(F_p - V|\psi_0|^2(\bar{\psi}_0 + \psi_0)\right)}. \qquad (13)$$

This expression reports how a perturbation changes the occupation of the pump state, which, for finite pump momentum, changes the overall current. To see this in detail, one can start with the time-independent Gross–Pitaevskii equation (GPE) for a bosonic field $\psi$ coupled to a current through the perturbation $\mathbf{f}(\mathbf{x})$:

$$\left(-\frac{\nabla^2}{2m} - \omega_p - i\kappa + V|\psi(\mathbf{x})|^2\right)\psi(\mathbf{x}) = -F_p e^{i\mathbf{k}_p \cdot \mathbf{x}}$$
$$-\frac{i}{2m}[\mathbf{f}(\mathbf{x}).\nabla\psi(\mathbf{x}) + \nabla.(\mathbf{f}(\mathbf{x})\psi(\mathbf{x}))], \qquad (14)$$

and find how the occupation of the pump state changes due to this perturbation. Because we are interested in the $\mathbf{q} \to \mathbf{0}$ limit, we take $\mathbf{f}$ to be independent of position, so this equation reduces to

$$\left(-\frac{\nabla^2}{2m} - \omega_p - i\kappa + V|\psi(\mathbf{x})|^2\right)\psi(\mathbf{x}) = -F_p e^{i\mathbf{k}_p \cdot \mathbf{x}} - \mathbf{f}.\frac{i\nabla}{m}\psi(\mathbf{x}). \qquad (15)$$

Rewriting $\psi(\mathbf{x}) = (\psi_0 + \phi(\mathbf{x}))e^{i\mathbf{k}_p \cdot \mathbf{x}}$, where $\psi_0 e^{i\mathbf{k}_p \cdot \mathbf{x}}$ is the mean-field satisfying the original GPE (i.e. with $\mathbf{f} = \mathbf{0}$), and removing any terms higher than first order in $\phi(\mathbf{x})$ and $\mathbf{f}$ leads to the equation

$$-\frac{\nabla^2}{2m}\phi(\mathbf{x}) - i\mathbf{k}_p.\frac{\nabla}{m}\phi(\mathbf{x}) + V\psi_0^2\phi^*(\mathbf{x})$$
$$+\left(\frac{k_p^2}{2m} - \omega_p - i\kappa + 2V|\psi_0|^2\right)\phi(\mathbf{x}) = \frac{\mathbf{f}.\mathbf{k}_p}{m}\psi_0. \qquad (16)$$

One may check (e.g. by Fourier transforming) that the solution of this is a homogenous (position independent) function $\phi(\mathbf{x}) = \phi$. The current associated with this change of occupation is given by $\delta\mathbf{j} = \left(\mathbf{k}_p/m\right)\left(\psi_0^*\phi + \phi^*\psi_0\right)$, which can be used to recover the mean-field response function through $\delta j_i = \chi_{ij}^{(0)}(\mathbf{q} \to \mathbf{0})f_j$, with $\chi_{ij}^{(0)}(\mathbf{q} \to \mathbf{0})$ as given in Eq. (13).

Figure 4 shows a quantity $R$ defined as the ratio of the long wavelength response to the mean-field density, $R = m^*\chi_{xx}(\mathbf{q} \to \mathbf{0})/|\psi_0|^2$. We explore how $R$ changes with pump momentum $k_p$ and mean-field polariton density $|\psi_0|^2$ for the parameters: $m^* = 5 \times 10^{-5}\,m_e$, where $m_e$ is the electron mass, $V = 0.01$ meV $\mu m^2$, and $\kappa = 0.05$ meV. The ratio increases quadratically with $k_p$, i.e. the higher the velocity of the pump state, the more the current changes due to the perturbation. That the response increases smoothly with velocity is consistent with a smooth increase in

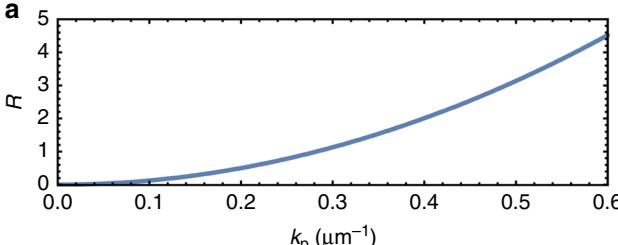

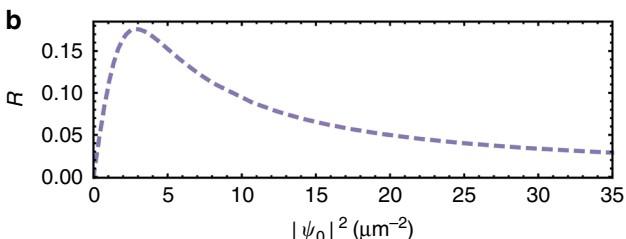

**Fig. 4** Response at zero detuning. The long wavelength response, normalised to the mean-field density, $R = m^{*}\chi_{xx}(\mathbf{q} \to \mathbf{0})/|\psi_{0}|^{2}$, as a function **a** of pump momentum $k_{p}$ at $|\psi_{0}|^{2} = 6.9\,\mu m^{-2}$, and **b** the mean-field density $|\psi_{0}|^{2}$ at $k_{p} = 0.1\,\mu m^{-1}$. The detuning is zero in both cases

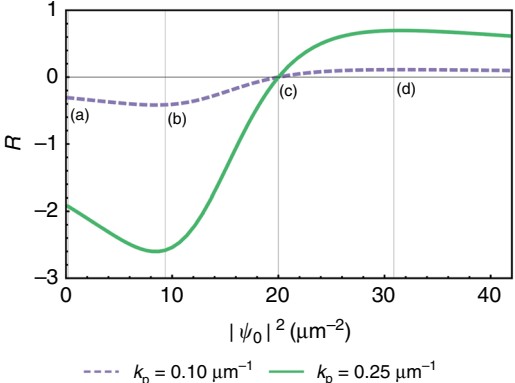

$$--- \quad k_{p} = 0.10\,\mu m^{-1} \qquad --- \quad k_{p} = 0.25\,\mu m^{-1}$$

**Fig. 5** Response at finite detuning. Total long wavelength response for $k_{p} = 0.1\,\mu m^{-1}$ (dashed purple) and $k_{p} = 0.25\,\mu m^{-1}$ (solid green) normalised with respect to the mean-field density, $R = m^{*}\chi_{xx}(\mathbf{q} \to \mathbf{0})/|\psi_{0}|^{2}$. Vertical lines marked (**a**–**d**) correspond to the excitation spectra in Fig. 2. For **a** and **b**, the response is negative, and goes to zero close to **c** where the excitation spectrum is of the linear Bogoliubov form (the mean-field component, Eq. (13), is exactly zero at **c** on account of an imaginary mean-field, but the fluctuations are small and finite), before becoming positive, peaking around **d**, and tailing off similarly to the resonant case in Fig. 4. Larger $k_{p}$ leads to a larger response to perturbations. Note that there is no change in response associated with the speed of sound, given by $c_{s} = \sqrt{V|\psi_{0}|^{2}/m^{*}}$, which corresponds to the momentum $k = 0.1\,\mu m^{-1}$ when $|\psi_{0}|^{2} = 6.1\,\mu m^{-2}$ and $k = 0.25\,\mu m^{-1}$ when $|\psi_{0}|^{2} = 38.1\,\mu m^{-2}$

drag force identified in a previous study[19] of coherently pumped polaritons. There is a nonmonotonic dependence of $R$ on the pump intensity, with an asymptotic decrease of $R$ at large intensity. This arises because a large pump intensity reduces the response to a weak perturbation.

**Detuning**. It is worth noting that blue detuning can lead to the real part of the excitation spectrum fulfilling the Landau criterion at the right density[28,35,40] (see Fig. 2(c)). This regime has been used to explore flows against defects with reduced dissipation[28]. Figure 5 shows how the long wavelength response normalised to

the mean-field density changes when we change the value of the polariton density for a specific blue detuning, where $m^{*} = 5 \times 10^{-5}\,m_{e}$, $V = 0.0025\,meV\mu m^{2}$, and $\kappa = 0.05\,meV$. It is notable that at the shifted resonance point, $\Delta_{p} = \omega_{p} - \varepsilon_{0} - V|\psi_{0}|^{2} = 0$, where the real part of the excitation spectrum takes a Bogoliubov form, the mean-field term in the response function, Eq. (13), becomes zero. The reason for this is that Eq. (13) is proportional to $\Re(\psi_{0})$, and when $\Delta_{p} = 0$, $\psi_{0}$ is entirely imaginary. While the long wavelength response due to fluctuations is finite, it is orders of magnitude smaller than the mean-field contribution at most densities. This suggests that, even if we take higher orders in fluctuations, there is always a pump strength for which the response goes from negative to positive and is thus strictly zero. Furthermore, this pump strength is very close to the shifted resonance point at which dissipationless flow was observed in experiments. For densities above the shifted resonance point, the Landau criterion is still fulfilled in the real part which is now gapped, but by contrast to the experimentally investigated regime in Fig. 2(c), the response function is finite and positive, showing a gradual reduction with increasing density similar to the case of zero detuning (Fig. 4).

Because there is no continuity of current (Eq. 3), the f-sum rule does not hold and there is no clear physical correspondence between the response functions and the density of the system. As a result, the negative value of the long wavelength response function at low densities does not present a problem. Additionally, it should be noted that in a previous study of incoherently pumped polaritons at the mean-field level[11], in which external potentials were present, it was concluded that the resultant currents in the steady state can change the physical picture substantially and render the interpretation of the superfluid and normal density fractions in terms of the response of the system to a vector potential unphysical. While the phase was free in that study and in the present work there are no external potentials, parallels exist with the fact that the coherent pumping ensures a steady-state current.

## Discussion

We study, using a nonequilibrium path integral method, a system of coherently pumped microcavity-polaritons in which the pumping is continuous, homogeneous, and below the OPO threshold, and show that the external fixing of the the macroscopic phase prevents it from being a superfluid. This is because the gapped spectrum affects the limiting behaviour of the current–current response function such that the longitudinal and transverse responses are the same. Remarkably, we also find that the system does not respond to either longitudinal or transverse perturbations at $\mathbf{k}_{p} = 0$, and it possesses only a normal component at finite $\mathbf{k}_{p}$, which grows with increasing $\mathbf{k}_{p}$. Rather than a superfluid, this result suggests the existence of a rigid state that, like a solid, has density but no current response. The smooth growth of the normal component with $\mathbf{k}_{p}$ is similar to the smooth crossover in drag predicted in a previous study[19]. Additionally, the fraction of the system corresponding to the macroscopic rigid state grows faster than that of the normal component as pump intensity is increased.

While blue detuning can allow the fulfilment of the Landau criterion in the real part, and signs of dissipationless flow have been observed[28], this is the only property associated with superfluidity that is exhibited by the rigid state, as vortices and persistent currents cannot form when the phase is externally fixed, and the superfluid response is zero. It is notable too that the long wavelength total current–current response function falls to zero very close to the experimentally investigated regime where the excitation spectrum takes the Bogoliubov form, which could explain the observed reduced scattering.

The existence of this rigid state suggests that driven-dissipative systems allow for a richer collection of macroscopic flow properties than equilibrium systems, and highlights the subtleties inherent in the attributes together known as superfluidity.

## Method

**Keldysh path integral.** We calculate the current–current response function using a Keldysh path integral technique[9,41,46]. Starting from the Hamiltonian (Eq. (1)) and integrating out the decay bath, the action is given in terms of Keldysh 'classical' and 'quantum' fields, $\Psi = (\psi^c, \psi^q)$[39]:

$$S[\Psi] = \sum_{\omega,\mathbf{k}} \left( \bar{\psi}^c_{\mathbf{k}}(\omega), \bar{\psi}^q_{\mathbf{k}}(\omega) \right) \begin{pmatrix} 0 & D^0_{\mathbf{k}}(\omega) \\ D^0_{\mathbf{k}}(\omega)^* & 2i\kappa \end{pmatrix} \begin{pmatrix} \psi^c_{\mathbf{k}}(\omega) \\ \psi^q_{\mathbf{k}}(\omega) \end{pmatrix}$$
$$- \sum_{\omega,\omega',\nu} \sum_{\mathbf{k},\mathbf{k}',\mathbf{q}} \frac{V}{2} \left( \bar{\psi}^c_{\mathbf{k}-\mathbf{q}}(\omega - \nu) \bar{\psi}^q_{\mathbf{k}'+\mathbf{q}}(\omega' + \nu) [\psi^c_{\mathbf{k}}(\omega) \psi^c_{\mathbf{k}'}(\omega')$$
$$+ \psi^q_{\mathbf{k}}(\omega) \psi^q_{\mathbf{k}'}(\omega')] + \text{c.c.} \right) - F_p(\bar{\psi}^q_0(0) + \psi^q_0(0)), \qquad (17)$$

where $D^0_{\mathbf{k}}(\omega) = \omega + \omega_p - \varepsilon_{\mathbf{k}} - i\kappa$ and $\varepsilon_{\mathbf{k}} = (\mathbf{k} + \mathbf{k}_p)^2/2\,m^*$. To connect this to the normal ordered current response, we add an extra term to the action, $\delta S$, that contains two source fields, $\mathbf{f}$ and $\boldsymbol{\theta}$, where the former is coupled to the Keldysh 'quantum' current and the latter to the observable current:

$$\delta S[\mathbf{f}, \boldsymbol{\theta}] = \sum_{\omega,\mathbf{k},\mathbf{q}} \gamma_i(\mathbf{k}+\mathbf{q}, \mathbf{k}) \bar{\Psi}_{\mathbf{k}+\mathbf{q}}(\omega) [\hat{\sigma}^K_x f_i(\mathbf{q}) + (\hat{\sigma}^K_z + i\hat{\sigma}^K_y) \theta_i(\mathbf{q})] \Psi_{\mathbf{k}}(\omega), \qquad (18)$$

where $\sigma^K_i$ are the Pauli matrices in the Keldysh basis.

**Current–current response function.** Having constructed a path integral, $\mathcal{Z} = \int \mathcal{D}(\bar{\Psi}, \Psi) e^{(iS + i\delta S)}$, the response function is found by differentiating it with respect to the source fields:

$$\chi_{ij}(\mathbf{q}) = -\frac{i}{2} \frac{d^2 \mathcal{Z}[\mathbf{f}, \boldsymbol{\theta}]}{df_i(\mathbf{q}) d\theta_j(-\mathbf{q})} \bigg|_{\mathbf{f}=\boldsymbol{\theta}=0}. \qquad (19)$$

Owing to the interaction term in the Hamiltonian, this calculation requires that we make the substitution $\Psi = \Psi_0 + \delta\Psi$ for a mean-field and quadratic fluctuations, modifying our path integral,

$$\mathcal{Z} = \int \mathcal{D}(\delta\bar{\Psi}, \delta\Psi) \exp\left[iS_0 + i\sum \delta\bar{\Psi}(\mathcal{D}^{-1} + A[\mathbf{f}, \boldsymbol{\theta}])\delta\Psi\right], \qquad (20)$$

where $\mathcal{D}^{-1}$ is the inverse matrix of Green's functions and $A[\mathbf{f}, \boldsymbol{\theta}]$ consists of the fluctuation terms dependent on the source fields. In general, our mean-field will be dependent on the source fields, and integrating out the fluctuations we find our response function is given by

$$\chi_{ij}(\mathbf{q}) = \chi^{\text{mf}}_{ij}(\mathbf{q}) + \chi^{\text{fl}}_{ij}(\mathbf{q}) \qquad (21)$$

$$= -\frac{i}{2} \left[ i \frac{d^2 S_0}{df_i(\mathbf{q}) d\theta_j(-\mathbf{q})} \right.$$
$$+ \frac{1}{2} \text{Tr}\left( \mathcal{D} \frac{dA}{df_i(\mathbf{q})} \mathcal{D} \frac{dA}{d\theta_j(-\mathbf{q})} \right)$$
$$\left. - \frac{1}{2} \text{Tr}\left( \mathcal{D} \frac{d^2 A}{df_i(\mathbf{q}) d\theta_j(-\mathbf{q})} \right) \right], \qquad (22)$$

where the first term comes from the mean-field and the others from the fluctuations.

**Coefficients.** The coefficients in Eqs. (7–10) are given in Supplementary Note 1.

## Data availability

Data sharing is not applicable to this article as no datasets were generated or analysed during the current study.

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

## Acknowledgements

We would like to thank I. Carusotto and K. Dunnett for helpful discussions, and A. Zamora for creating Fig. 1(a). M.H.S. acknowledges financial support from EPSRC (Grants no. EP/I028900/2 and no. EP/K003623/2) and J.K. from EPSRC program Hybrid Polaritonics (EP/M025330/1).

## Author contributions

R.T.J. performed the analytical and numerical calculations and J.K. and M.H.S. suggested the problem and supervised the work. All authors contributed to writing the paper.

## Additional information

**Competing interests:** The authors declare no competing interests.

