## [Peer Review File · Nature Communications]

Reviewers' comments:

Reviewer #1 (Remarks to the Author):

The authors consider coherently driven microcavity polaritons, study their properties, and, in particular, attempt to answer whether it forms a superfluid state or not. Calculating the superfluid response function via the Keldysh field theory, they conclude that the steady state of the driven-dissipative model is not a superfluid. Nevertheless, they find that, when the coherent drive is coupled to the system at zero momentum, the system forms a rigid state that is neither a superfluid nor a normal fluid.

The authors argue that, while the real part of the excitation spectrum fulfills the Landau criterion, it is essential to include the imaginary part as well. The latter, however, is always gapped. This observation is key to the absence of superfluid response. This argument makes sense to me, but also makes the result rather trivial: if the system is not fully gapless (both real and imaginary parts), it cannot possibly be a superfluid.

The emergence of the rigid state is interesting. Nevertheless, this state is inspected only at the mean-field level as the authors argue that the contribution due to fluctuations are orders of magnitude smaller. However, it would be important to see whether this result holds in the thermodynamic limit or not, that is, whether it forms a new phase.

Before I commit myself to a decision, I'd like to ask the authors a number of questions in order that they can further clarify their key results.

- The coherent drive, by explicitly breaking the symmetry, is almost bound to make the system gapped. While the detailed diagrammatic calculations determine the gap and the response function, should it not be expected that the resulting state is gapped, and hence not a superfluid?
- The model considered here bears close resemblance to driven-dissipative Bose-Hubbard model which has been studied extensively in the literature. Examples are Phys. Rev. Lett. 110, 233601 (2013) and Phys. Rev. A 95, 043826 (2017). It is argued that at the critical point, these models exhibit an Ising critical behavior that is quite distinct from a superfluid phase. How do the authors compare their results against those in coherently driven-dissipative Bose-Hubbard systems?
- The real part of the excitation spectrum satisfies the Landau criterion to the lowest-order. Do fluctuation corrections spoil this feature?
- A somewhat similar question for the rigid state: does it survive fluctuations? If not, shouldn't it be merely interpreted as a rather "stiff" liquid? Also what is the length scale beyond which the rigid state behaves like a liquid? And, how does this length scale compare to that of the emergence of KPZ physics?
- There seems to be a typo in defining α^{\pm} (they are defined identically).

Reviewer #3 (Remarks to the Author):

The paper "Can coherently driven microcavity exciton-polaritons be a superfluid?" by R.T. Jiggins et al. is devoted to the theoretical study of superfluid response, measured in terms of the difference between the responses to longitudinal and transverse fields, in a coherently driven exciton-polariton system. The main results of the paper are that such defined superfluid response is zero. In spite of the previously demonstrated absence of scattering, it is shown that these systems do not display another important characteristic of a superfluid. The authors demonstrate instead the existence of a rigid state which does not respond to small perturbations, and suggest that the notion of superfluidity should be reexamined in the case of driven-dissipative system.

In my opinion, the presented results will be of high interest to the theoreticians working on superfluidity or superconductivity of open systems. These results can be a milestone in the

understanding of these systems, and deserve to be published in a high-visibility journal. However, in my opinion the current style of presentation of the results does not meet the specific criteria of Nature Communications, and the paper could be possibly misleading. To be more specific:

- I believe that unfortunately the core of the paper, which describes the method of calculation of the responses using the Keldysh theory will not be comprehensible to a non-specialist. This method is rather abstract and mathematical, and the authors should put more effort to make it understandable. In particular, the sections about sum rules and response functions are very hard to follow and unclear. For example, no clear physical interpretation of the current operator χ has been provided.
- The authors do not explain the importance of their calculations clearly enough. As I understand they conclude that in driven-dissipative systems the response is no longer related to the density of the superfluid or normal fractions and can even become negative. In this case, one can ask what is the new physical interpretation of the response, and is it really so fundamentally important to calculate it? Or if it has no clear physical interpretation, shouldn't we rather conclude that the absence of scattering, which has very clear physical consequences, is more fundamental in open systems?
- As the authors write in the introduction, the fundamental property of the superfluid is that the bulk of the system is irrotational. In the coherently driven system, the rigid state is also irrotational, so this does not seem to be against superfluidity.
- What may be misleading in the paper is that the authors investigate only the coherently driven system which is locked by an external laser. However there is much experimental evidence that superfluidity, solitons, and vortices can exist in coherently driven system in the area beyond the pumping spot, which is not phase locked. The authors overlook this case but it may be an example of a decaying superfluid, with the lifetime given by the polariton lifetime. In principle, the authors method could be suitable to investigate this case as well. I am quite curious if the superfluid response could be finite in this case.
- there is a mistake in the definition of α^{\pm} under Eq (2).

Response to referees

We would like to thank the referees for such a careful reading of our manuscript and for their constructive comments aimed at making it clearer and more accessible to non-specialists. Below we respond to all the comments in detail and indicate how we have changed the manuscript in response to those comments.

Reviewer #1 (Remarks to the Author):

The authors argue that, while the real part of the real part of the excitation spectrum fulfils the Landau criterion, it is essential to include the imaginary part as well. The latter, however, is always gapped. This observation is key to the absence of superfluid response. This argument makes sense to me, but also makes the result rather trivial: if the system is not fully gapless (both real and imaginary parts), it cannot possibly be a superfluid.

• The coherent drive, by explicitly breaking the symmetry, is almost bound to make the system gapped. While the detailed diagrammatic calculations determine the gap and the response function, should it not be expected that the resulting state is gapped, and hence not a superfluid?

While we fully agree with the referee that the matter might be clear or even obvious after reading our paper, it evidently was not clear to the community given the claims which have previously been made in high profile papers. The idea that a system with a gap in the imaginary part of the spectrum cannot be a superfluid is not widely recognised, and indeed there exist claims contradicting this (for example Amo et al. Nature Phys. 5 (2009)). Part of the motivation of our work is to change this widespread perception, and to achieve this we do not believe it is sufficient to simply make a general argument – rather, it is necessary to actually calculate the superfluid response of the system and show beyond doubt that the gapped spectrum prevents superfluidity.

The emergence of the rigid state is interesting. Nevertheless, this state is inspected only at the mean-field level as the authors argue that the contribution due to fluctuations are orders of magnitude smaller. However, it would be important to see whether this result holds in the thermodynamic limit or not, that is, whether it forms a new phase.

Given this comment we believe we were not clear in the manuscript about some of the details. Indeed, we compute *both* the mean-field and the fluctuation parts of the response (not only the mean-field part). We find (after doing the calculations) that for relevant parameters, the part of the response coming from fluctuations is orders of magnitude smaller than the response coming from the mean-field. That is why in figure 3 of the old version of the paper we presented only the mean-field response. However, to avoid confusion we have now calculated the fluctuation corrections to this mean-field response and in the current version (now in figure 4) we have plotted the total response (combining both the mean-field and the fluctuation parts).

Regarding the general question of whether fluctuation corrections have a significant effect, it is worth noting the fact that because the system we consider is gapped, the effect of fluctuations is generally a finite correction. We discuss this further below, in response to the referee's comment on the KPZ nonlinearity. In the manuscript we have added a sentence: "Because the excitation spectrum is gapped, the KPZ nonlinearity is not relevant and our result holds in the thermodynamic limit". Note that due to the gap there is a mass-like term in the equation for fluctuations, which

means they are energetically costly. The state is robust to these fluctuations, which we find to be small. It is important to clarify that the KPZ equation for the phase exists only for systems where there is no mass term for the phase, in which case fluctuations can be easily excited. While the gap precludes superfluidity it also stabilises the state against fluctuations making the rigid state valid in the thermodynamic limit.

• **The model considered here bears close resemblance to driven-dissipative Bose-Hubbard model which has been studied extensively in the literature. Examples are Phys. Rev. Lett. 110, 233601 (2013) and Phys. Rev. A 95, 043826 (2017). It is argued that at the critical point, these models exhibit an Ising critical behavior that is quite distinct from a superfluid phase. How do the authors compare their results against those in coherently driven-dissipative Bose-Hubbard systems?**

There exists an important distinction between the scenario we consider in our manuscript, and that considered in these other works. For a resonantly pumped nonlinear optical or polariton mode, if the pump is blue-detuned with respect to the bare dispersion, the system will exhibit two stable states for some range of pump powers: a low occupation ‘dim’ state and a high occupation ‘bright’ state. This bistability does not exist for all powers as there is a lower critical power set by the cavity loss.

The two papers mentioned by the referee regarding the Bose-Hubbard model are largely concerned with this bistability, and the phase diagram resulting from this. More explicitly, Phys. Rev. Lett. 110, 233601 (2013) looks at the phase diagram of an array of coupled cavities and finds regions distinguished by the number of stable states and Phys. Rev. A 95, 043826 (2017) shows a mapping between bistability on a lattice and a classical Ising model. Our study, by contrast, is concerned only with the monostable state at low pump intensity. This is because that is the regime that was investigated by experiments which were looking for superfluidity (Amo et al. Nature Phys. 5 (2009)). We have emphasised this fact by adding a comment in the paper.

Additionally, to our knowledge no investigation into the current-current response function has been made in the Bose-Hubbard case, so we cannot directly compare our work to these other papers.

• **The real part of the excitation spectrum satisfies the Landau criterion to the lowest-order. Do fluctuation corrections spoil this feature?**

The effect of fluctuation corrections will be to shift the energy at which the real part satisfies the criterion. Indeed, one can see from figure 2 that the branches of the real part cross over in (a) and are gapped at (d), meaning that there must always be a detuning and pump power where the branches touch at a single point. Thus including fluctuations will simply shift the pump power at which the real part satisfies the Landau criterion. Furthermore, as noted above, in our system the effect of the fluctuations is small.

• **A somewhat similar question for the rigid state: does it survive fluctuations? If not, shouldn't it be merely interpreted as a rather “stiff” liquid? Also what is the length scale beyond which the rigid state behaves like a liquid? And, how does this length scale compare to that of the emergence of KPZ physics?**

The rigid state does indeed survive fluctuations. As we have mentioned above, the issue regarding the KPZ nonlinearity in Altman et al. PRX 5 (2015), Sieberer et al. PRB 94 (2016), Wachtel et al. PRB

94 (2016)) pertains to the incoherently pumped system where the excitation spectrum is gapless. In our case, there is a mass term which dominates over the KPZ nonlinearity (which is proportional to ∇^2) in the long distance limit. Thus our results hold in the thermodynamic limit. We have added a comment at the end of the introduction emphasising this point.

Previously figure 3 (now figure 4) showed the ratio between the mean field response and the mean field density. We have now redone this calculation including fluctuations. As is mentioned in the text, the fluctuation parts are orders of magnitude smaller than the mean field parts, so the figure looks identical. However, in light of the referee's comment, we think it is important to be as clear as possible that our results include fluctuations and our conclusions are robust to them.

- **There seems to be a typo in defining α^{μ} (they are defined identically).**

We thank the referee for spotting this. The mistake has been fixed: μ has been added in front of the momentum argument in the dispersion.

Reviewer #3 (Remarks to the Author):

- I believe that unfortunately the core of the paper, which describes the method of calculation of the responses using the Keldysh theory will not be comprehensible to a non-specialist. This method is rather abstract and mathematical, and the authors should put more effort to make it understandable. In particular, the sections about sum rules and response functions are very hard to follow and unclear. For example, no clear physical interpretation of the current operator χ has been provided.

This is a very important point and we agree that there is more that can be done to make our work clear to a general reader.

In the section titled 'Sum-rules and anisotropy' we have tried to give a more physical description of χ , including a new figure (figure 2) explaining what is meant by longitudinal and transverse response. We have also included the Hamiltonian for the perturbation and emphasised that the f-sum rule is really just a statement of the continuity of current.

In the 'Response function' section we have made an effort to clarify that the material presented in the main text is now just our results, and that the derivation is restricted to the methods. We hope that by being clear about this separation our results section will come across as self-contained and less technical.

We have also re-arranged how we present the response function to further improve clarity. Previously, we wrote three separate contributions to the response function, corresponding to how terms arise from the field theoretic calculation. We have now re-arranged our calculation to distinguish only mean-field and fluctuation parts of the result, as these have a more obvious direct physical interpretation.

Finally, we have clarified exactly how one defines the various Green's functions we reference as we are aware this is not common knowledge. Overall, we have tried to cut down on the technical language at the start of the response functions section in favour of a more descriptive approach.

- The authors do not explain the importance of their calculations clearly enough. As I understand they conclude that in driven-dissipative systems the response is no longer related to the density of the superfluid or normal fractions and can even become negative. In this case, one can ask what is the new physical interpretation of the response, and is it really so fundamentally important to calculate it? Or if it has no clear physical interpretation, shouldn't we rather conclude that the absence of scattering, which has very clear physical consequences, is more fundamental in open systems?

The importance of calculating the response is that it provides the best metric for defining what is 'super' about superfluidity. Given that all real superfluids are at finite temperature, they will always have a viscous normal component. As such, a real superfluid (i.e. not at zero temperature) does not have vanishing viscosity. This means a superfluid at finite temperature with reduced but *finite* viscosity cannot be sharply distinguished from other merely low viscosity fluids by this definition alone. In contrast, the definition based on the response functions allows a clear and unambiguous distinction between a low viscosity normal fluid and a superfluid. Additionally, the 'super' in superfluidity means more than low viscosity. It is a specific, quantum mechanical phenomenon that has a further series of characteristics: (bulk) irrotational flow, quantised vortices, and metastable persistent currents. This is very much analogous to stating that a superconductor is not merely defined by low electrical resistance – instead it is the Meissner effect (which depends on the charged particle analogue of the response function) which *defines* superconductivity.

All of the above properties of a superfluid can be found to follow from the difference between the longitudinal and transverse responses. Both the superfluid and normal components respond to longitudinal perturbations and (if current is conserved) this provides a measure of the total density of the system. The transverse response, by contrast, gives a measure just of the normal component and putting these all together allows one to compute the superfluid and normal fractions of the system. These fractions can be experimentally measured by looking at the moment of inertia of a superfluid system and found to match the theoretical values.

The lack of transverse response in the superfluid case derives from the macroscopic wavefunction, from which the existence of vortices and persistent currents can also be inferred. Furthermore, calculating the superfluid fraction provides a framework for thinking about Landau's two-fluid model and the existence of a small amount of scattering at finite temperatures.

In our case, current is not conserved, and so we cannot directly relate the response function to the density. Nevertheless, we can still find whether any part of the system behaves as a superfluid and show that for the case we consider, there is no such superfluid part. That the response is sometimes negative does not change the fact that there is no superfluid response. Moreover, the response function is allowed to be negative given the violation of current continuity. We have been clearer about this in the paragraph preceding the start of the 'Discussion'.

- As the authors write in the introduction, the fundamental property of the superfluid is that the bulk of the system is irrotational. In the coherently driven system, the rigid state is also irrotational, so this does not seem to be against superfluidity.

There is a critical difference between a superfluid and the rigid state – while neither respond to transverse perturbations, unlike a superfluid the rigid state does not respond to longitudinal

perturbations either. In this sense, it is more like a solid than a fluid (albeit without any other properties of solids). Certainly, we would not want to call other rigid bodies a superfluid.

- What may be misleading in the paper is that the authors investigate only the coherently driven system which is locked by an external laser. However there is much experimental evidence that superfluidity, solitons, and vortices can exist in coherently driven system in the area beyond the pumping spot, which is not phase locked. The authors overlook this case but it may be an example of a decaying superfluid, with the lifetime given by the polariton lifetime. In principle, the authors method could be suitable to investigate this case as well. I am quite curious if the superfluid response could be finite in this case.

Looking beyond the pump spot is beyond the scope of our work – indeed we concentrate on the most transparent case of a uniform pump which was the regime investigated experimentally in Amo et al. Nature Phys. 5 (2009). The response function in the absence of coherent driving has been investigated before (Keeling PRL 107 (2011), Keeling et al. Universal Themes of Bose-Einstein Condensation (2017)) and it was found that superfluidity can survive in the presence of dissipation (although the later discovery of the effect of the KPZ nonlinearity modifies this story somewhat).

With regards to the experimental evidence of vortices and solitons (Amo et al. Science 332 (2011), Nardin et al. Nature Phys. 7 (2011)) we have added in the first paragraph of the ‘Rigid state’ section a reference to these experiments, along with an explanation that these systems are different to the one we study as the restrictions they place on pump, either because it is pulsed or spatially finite, relax the phase fixing.

We have also explained more clearly at the end of the introduction what we mean by ‘coherently’ pumped – that is, we are specifically concerned with a continuously and homogeneously pumped system below the OPO threshold.

- there is a mistake in the definition of α^{\pm} under Eq (2).

We thank the referee for spotting this. The mistake has been fixed: \pm has been added in front of the momentum argument in the dispersion.

Reviewers' comments:

Reviewer #1 (Remarks to the Author):

The authors have provided satisfactory explanations in most of their response to my previous comments. There is one point which however needs further clarification. For a rigid state, like a solid, the system's response to perturbations is strictly zero. Even if this response is small, as long as it is finite, the system is not strictly rigid. To further clarify this point, let me invoke a familiar example: a continuous symmetry cannot be broken in two dimensions at finite temperature due to Mermin-Wagner theorem. Hence, a solid in two dimensions cannot exist at finite temperature; however, there could exist a length scale below which there is an ordered solid phase. In a similar sense, the rigid "phase" is not really a phase in the thermodynamic limit. Upon clarifying this point, I think this paper can be published in Nature Communications.

Reviewer #3 (Remarks to the Author):

The new version of the manuscript takes into account my suggestions. In particular, the section on sum rules and response functions is now much more clear than in the previous version. There are still parts of the text, especially in the following section, which will not be readable for a non-specialist. But I believe that the authors put appropriate effort in making their ideas as clear as possible.

I find the response to the other points of my report satisfactory. In my opinion the manuscript has been improved considerably. Given the importance of the results to the fields of superfluidity and superconductivity, I recommend publication of the manuscript in the present form.

Second referees response

Reviewer #1 (Remarks to the Author):

The authors have provided satisfactory explanations in most of their response to my previous comments. There is one point which however needs further clarification. For a rigid state, like a solid, the system's response to perturbations is strictly zero. Even if this response is small, as long as it is finite, the system is not strictly rigid. To further clarify this point, let me invoke a familiar example: a continuous symmetry cannot be broken in two dimensions at finite temperature due to Mermin-Wagner theorem. Hence, a solid in two dimensions cannot exist at finite temperature; however, there could exist a length scale below which there is an ordered solid phase. In a similar sense, the rigid "phase" is not really a phase in the thermodynamic limit. Upon clarifying this point, I think this paper can be published in Nature Communications.

We thank the referee for encouraging us to clarify this important point.

Regarding whether the response is strictly zero or just small, there is an important difference between the case where the pump is at zero wavevector, and the case with non-zero wavevector. When the pump is at zero wavevector, the response is indeed strictly zero, and so, as the referee notes, this case can be called strictly rigid. When the pump is at non-zero wavevector, there is a non-zero response, however even in this case, this is not a conventional 'normal' response of a fluid. For a pump at non-zero wavevector, one can vary the net current by varying the amplitude of the coherent state, which is possible since polariton number is not conserved. This means that when a force is applied, an extra current can occur purely through a change in the density of the condensate, rather than a change in the momentum of the occupied states. In this sense, the state remains rigid, locked to the pump laser.

There is however an exception to this non-zero response when the pump is blue detuned, where there is always a unique pump strength for which the response at finite wavevector including fluctuations is strictly zero. This follows from the fact that the total response goes from negative to positive as a function of pump power, so there will always be a power for which it is zero. Taking higher orders of fluctuations might shift the exact position of this point but it will still exist. Notably, this point coincides very closely with when the real part of the excitation spectrum is of Bogoliubov form, which is the regime in which dissipationless flow has been experimentally observed.

While clearly the rigid state is not exactly the same as a solid, it bears enough resemblance to be in our opinion a useful analogy. We do however recognise that we may not have been clear enough about what we mean and have amended the start of the second paragraph of the 'Rigid state' section and the first paragraph of the 'Detuning' section in the paper to clarify the points discussed above.

Since the referee mentions the analogy of a 2D solid and the Mermin-Wagner theorem, we note also the relation between our results and this theorem. Importantly, our rigid state does not involve spontaneous symmetry breaking, as the existence of the coherent pump means there is no symmetry to start with. The phase is fixed externally and the Mermin-Wagner theorem does not apply.